# Pre-Treatment HIV Drug Resistance and Genetic Diversity in Cameroon: Implications for First-Line Regimens

**DOI:** 10.3390/v15071458

**Published:** 2023-06-28

**Authors:** Joseph Fokam, Collins Ambe Chenwi, Valère Tala, Désiré Takou, Maria Mercedes Santoro, George Teto, Beatrice Dambaya, Felix Anubodem, Ezechiel Ngoufack Jagni Semengue, Grace Beloumou, Sandrine Djupsa, Edgar Assomo, Charles Fokunang, Claudia Alteri, Serge Billong, Nounouce Pamen Bouba, Rogers Ajeh, Vittorio Colizzi, Dora Mbanya, Francesca Ceccherini-Silberstein, Carlo-Federico Perno, Alexis Ndjolo

**Affiliations:** 1Chantal BIYA International Reference Centre for Research on HIV/AIDS Prevention and Management, Messa, Yaoundé P.O. Box 3077, Cameroon; talavalre@yahoo.com (V.T.); dtakou@yahoo.com (D.T.); ggteto@yahoo.fr (G.T.); dbambaya@yahoo.fr (B.D.); anubodemfesong@yahoo.com (F.A.); ezechiel.semengue@gmail.com (E.N.J.S.); graceangong12@yahoo.fr (G.B.); djupsans@yahoo.fr (S.D.); huguesassomo@yahoo.fr (E.A.); colizzi@bio.uniroma2.it (V.C.); cf.perno@uniroma2.it (C.-F.P.); andjolo@yahoo.com (A.N.); 2Faculty of Health Science, University of Buea, Buea P.O. Box 0063, Cameroon; 3Faculty of Medicine and Biomedical Sciences, University of Yaoundé I, Yaoundé P.O. Box 1364, Cameroon; charlesfokunang@yahoo.co.uk (C.F.); serge.billong@cnls.cm (S.B.); dmbanya1@yahoo.co.uk (D.M.); 4National HIV Drug Resistance Prevention and Surveillance Working Group, Ministry of Public Health, Yaoundé P.O. Box 3038, Cameroon; 5Department of Experimental Medicine, Faculty of Medicine and Surgery, University of Rome “Tor Vergata”, Via Montpellier 1, 00133 Rome, Italy; santormaria@gmail.com (M.M.S.); ceccherini@med.uniroma2.it (F.C.-S.); 6Faculty of Sciences and Technologies, Evangelical University of Cameroon, Bandjoun P.O. Box 0127, Cameroon; 7Department of Oncology and Hemato-Oncology, University of Milan, Via Festa del Perdono 7, 20122 Milano, Italy; claudia.alteri@uniroma2.it; 8Department of Disease, Epidemic and Pandemic Control, Ministry of Public Health, Yaounde P.O. Box 3038, Cameroon; boubapamen@gmail.com; 9Central Technical Group, National AIDS Control Committee, Yaoundé P.O. Box 2005, Cameroon; ajehrogers@gmail.com; 10National Blood Transfusion Service, Ministry of Public Health, Yaoundé P.O. Box 3038, Cameroon; 11Haematology and Transfusion Service, Centre Hospitalier et Universitaire (CHU), Yaoundé P.O. Box 30335, Cameroon; 12Bambino Gesu’ Children’s Research Hospital, Piazza S. Onofrio 4, 00165 Rome, Italy

**Keywords:** pre-treatment drug resistance, HIV-1, genetic diversity, first-line regimens, Cameroon

## Abstract

The efficacy of first-line antiretroviral therapy (ART) may be hampered by the presence of HIV drug resistance (HIVDR). We described HIV-1 pre-treatment drug resistance (PDR) patterns, effect of viral clades on PDR, and programmatic implications on first-line regimens in Cameroon. A sentinel surveillance of PDR was conducted from 2014 to 2019. Sequencing of HIV-1 protease and reverse transcriptase was performed, and HIVDR was interpreted using Stanford HIVdb.v.9.4. In total, 379 sequences were obtained from participants (62% female, mean age 36 ± 10 years). The overall PDR rate was 15.0% [95% CI: 11.8–19.0] nationwide, with significant disparity between regions (*p* = 0.03). NNRTI PDR was highest (12.4%), of which 7.9% had DRMs to EFV/NVP. Two regions had EFV/NVP PDR above the 10% critical threshold, namely the Far North (15%) and East (10.9%). Eighteen viral strains were identified, predominated by CRF02_AG (65.4%), with no influence of genetic diversity PDR occurrence. TDF-3TC-DTG predictive efficacy was superior (98.4%) to TDF-3TC-EFV (92%), *p* < 0.0001. The overall high rate of PDR in Cameroon, not substantially affected by the wide HIV-1 genetic diversity, underscores the poor efficacy of EFV/NVP-based first-line ART nationwide, with major implications in two regions of the country. This supports the need for a rapid transition to NNRTI-sparing regimens, with TDF-3TC-DTG having optimal efficacy at the programmatic level.

## 1. Introduction

In 2018, 37.9 million people worldwide (70% of them in Africa) were living with the human immunodeficiency virus (HIV) [1]. New infections remain numerous, with 1.7 million people being newly infected during the same year [1]. Considerable progress has been made in the care of infected people, reducing mortality by 48% since the highest level in 2005 [1]. In fact, 800,000 people died from AIDS-related illnesses worldwide in 2018, compared to 1.9 million in 2005 and 1.5 million in 2010 [1]. In Africa, an estimated 25.7 million people were living with HIV in 2018, making this region the most affected in the world [1]. In Cameroon, around 540,000 people were living with HIV in the same year, 52% of whom were on antiretroviral therapy (ART) [2].

ART aims to control viral replication and ensure immune reconstitution, biologically translated by a suppression of the viral load and an increase in the CD4 T lymphocyte count [3]. Thus, antiretroviral molecules allow infected people to live longer and in good health. In countries with limited resources, the current first-line treatment consists of 2 identical nucleoside reverse transcriptase inhibitors (NRTIs) and 1 non-nucleoside reverse transcriptase inhibitor (NNRTI), the objective being viral suppression during the first year of initiation of ART [3]. According to the new WHO guidelines, ART is recommended for anyone infected with HIV regardless of the WHO clinical stage or CD4 count [3]. These efforts, with the joint United Nations program on HIV/AIDS (UNAIDS), have resulted in the therapeutic care of approximately 24.5 million people living with HIV worldwide (including approximately 16 million in sub-Saharan Africa) by mid-June 2019 [1]. This is further illustrated in our context by the increased number of patients accessing HIV related care, with 17.1 million in 2015 versus 7.7 million in 2010 [1]. In Cameroon, the ART coverage rate is currently estimated at 93.1%, with 80.1% on viral load suppression [4].

With the scale-up of ART, the risk of the emergence and transmission of HIV drug resistance is more of a public health concern. This resistance to HIV occurs when the virus replicates in the presence of one or more antiretrovirals (ARVs) [5]. It is caused by mutations in the genetic structure of HIV at the active site of ARV drugs, thereby altering the ability of ARVs to block viral replication. There are several types of HIVDR, of which: (i) acquired drug resistance that develops among patients failing ART; (ii) transmitted drug resistance that is detected in a newly infected patient, naive on ART with no notion of prior ARV exposure; and (iii) pre-treatment drug resistance (PDR) detected in a patient initiating ART, with or without prior ARV exposure [6]. To improve the effectiveness of treatment programs for people living with HIV, WHO recommends monitoring of HIVDR, be it transmitted or acquired [7]. Therefore, PDR surveys should generate nationally representative estimates of (1) the PDR among adults initiating ART, regardless of previous ARV drug exposure, (2) the prevalence of PDR among ARV drug–naive adults initiating ART, and (3) the proportion of adult ART initiators reporting previous ARV drug exposure [8]. For regions with a prevalence of resistance to NNRTI pre-treatment greater than or equal to 10%, the WHO recommends taking certain measures, namely: starting ART with Dolutegravir or choosing antiretroviral molecules on the basis of a genotypic resistance test [9]. According to the WHO, one in ten patients initiating ART harbor a virus carrying one or more resistance mutations, with women being two times more affected than men [10]. As described by Gupta et al., the threshold for PDR is generally increasing in low-income countries, especially in East Africa, but also in West and Central Africa, with an estimated annual increase of 3% [11]. In sub-Saharan Africa, the prevalence of transmitted resistance varies from 1.1% to 12.3% [12]. WHO estimates that a pre-treatment resistance to NNRTIs higher than 10% will be responsible for 105,000 new infections and 135,000 HIV-related deaths between 2016 and 2021 [13]. Studies in Cameroon showed an overall pre-treatment NNRTI resistance threshold of 8.1% [14]. However, evidence is lacking as it concerns geographic disparity across regions of the country, effects of subtypes on PDR and potentially active first-line ARVs.

In view of epidemiological surveillance, we proposed an evaluation of the genotypic resistance profile of PDR in patients initiating ART and its implication on Dolutegravir-based versus TDF-based ART regimensin the Cameroonian context. Specifically, we determined the national threshold for pre-treatment HIV-1 resistance and its variability across regions; the patterns of resistance mutations according to classes of antiretroviral molecules; different circulating viral subtypes and their effect on HIV-1 PDR; and finally, predicted effectiveness of the standard regimen of Tenofovir–Lamivudine–Dolutegravir (TLD) as compared to the alternative first-line regimen of Tenofovir–Lamivudine–Efavirenz (TLE) [15].

## 2. Materials and Methods

### 2.1. Study Design, Sites, and Population

We carried out a cross-sectional and analytical study for five years, from December 2014 to June 2019. The enrollment of participants was carried out in health facilities of eight regions of Cameroon: Centre, Littoral, East, West, Northwest, Southwest, North, and Far North. In each region, patients were recruited in both urban and rural sites at the hospitals with the highest patient frequency. Table 1 shows the health facilities from which patients were recruited. HIV-1 genotypic resistance tests were performed at the Virology Laboratory of the Chantal Biya International Reference Center (CIRCB). The study participants consisted of adult patients initiating ART in the health facilities of regions previously cited. Using previous HIV prevalence studies in each region, considering a 5% error margin and 95% confidence interval, an estimated minimum sample size was calculated for each region, following which we based our enrollment.

### 2.2. Selection Criteria

Included in our study were HIV-1 infected patients aged above 19 years (adults per WHO definitions) with no prior history of ART treatment. Not included in our study were patients who were re-initiating ART, those whose samples were unsuccessful at sequencing, and those who were co-infected with either viral hepatitis B or C. Patients co-infected with viral hepatitis were to excluded to limit confounders related to potential HBV drug exposure that interferes with HIV PDR selection (tenofovir, lamivudine, emtricitabine); HCV/HIV co-infection also exhibits differences in viral genotypes as compared to mono-infection.

### 2.3. Data Collection

Patients presenting for ART initiation at the various recruitment sites were encountered by the recruitment clinician. Data collection was carried out using a pre-established data abstraction form at each site after obtaining a thorough medical history and clinical examination to minimize recruitment of participants with previous drug exposure. Data collected included the following: age, sex, region of origin, zone of residence (urban/rural), and WHO clinical stage. For each participant, 10 mL of venous blood was collected in an EDTA tube. The tubes were centrifuged at a speed of 1800 rpm, separated for plasma collection, plasma aliquoted (1 mL) in cryotubes, stored at −20 °C, and transported to the CIRCB, where biological analyzes were carried out.

### 2.4. Ethical Considerations

This study was conducted in accordance with the ethical principles of the 1964 Helsinki Declaration, revised in October 2013. Ethical clearance was obtained from the Institutional Research Ethics Committee (CIER) of the Faculty of Medicine and Biomedical Sciences, University of Yaoundé 1 (FMBS/UY1), as well as the authorizations of the different structures concerned by the study. Informed consent from all participants was obtained prior to inclusion after explaining the purpose, benefits, and possible risks of the study. Biological exams were free for all participants, and the results were given to the treating physicians to optimize the treatment.

### 2.5. Genotypic Resistance Test

Genotypic resistance testing was carried out in the pol gene (reverse transcriptase and protease regions) as follows. Viral RNA was extracted from 1 mL of plasma using the QIAGEN kit, following the manufacturer’s instructions. For the amplification of the pol region, a previously validated inhouse genotyping assay was used [16]. The target sequence was first amplified with Reverse Transcription PCR using BS primers (“5′-GAC AGG ATT ATT TTT TAG GG-3′”) and FRA S1 (“5′-TT CCC CAT ATT ACT ATG CTT-3′”) in 25 μL of reaction mixture for 40 cycles. The PCR conditions were 50 °C for 30 min, 94 °C for 2 min, 95 °C for 30 s, 51 °C for 30 s, and 72 °C for 2 min, followed by extension at 72 °C for 10 min. A semi-Nested PCR was subsequently performed using BS primers (“5′-GAC AGG ATT ATT TTT TAG GG-3′”) and TAK3 (“5’-GGC TCT TGA TAA TGA TAT TAT GT-3′”) in 50 μL of reaction mixture for 30 cycles. The PCR conditions were 93 °C for 12 min, 94 °C for 30 s, 53 °C for 45 s, and 72 °C for 2 min, followed by extension at 72 °C for 10 min. The PCR products were revealed using 1% agarose gel electrophoresis. The successfully amplified samples were purified and then sequenced by the following primers: B (“5′-AGC AGA CCA GAG CCA ACA GC-3′”), F (“5′-CCA TCC ATT CCT GGC TTT AAT-3′”), SEQ1 (“5′-GAA TGG ATG GCC CAA AA-3′”), SEQ2 (“5′-TTG AGA TAC AAT GGA AAA GGA AGG-3′”), SEQ3 (“5′-CCC TGT GGA AAG CAC ATT GTA-3′”), SEQ4 (“5′-GCT TCC ACA GGG ATG GAA-3′”), SEQ5 (“5′-CTA TTA AGT CTT TTG ATG GGT CA-3′”), and TAK3 (“5′-CCT TGT TTC TGT ATT TCT GCT-3′”). The products of the sequencing reaction obtained were purified by exclusion chromatography on SEPHADEX G50 resin and then sequenced using a genetic analyzer (ABI 3500^®^).

### 2.6. Sequence Analysis for HIVDR Profiling, Drug Efficacy, and Subtyping

The resulting sequences encompassing the pol region were assembled and edited manually using Seqscape software version 2.5 and Recall version 2.28 Resistance mutations were interpreted using Stanford HIVdb software version 9.4 (http://www.hivdb.stanford.edu, accessed on 24 May 2023), and all resistance-associated mutations in the samples were identified. The predictive efficacies of each antiretroviral molecule were calculated according to the genotypic drug susceptibility score provided according to HIVdb algorithm (http://www.hivdb.stanford.edu, accessed on 24 May 2023). The regimen efficacy for TLD versus TLE was calculated irrespective of the presence of M184V due to the beneficial effect of this mutation in improving TDF (a molecule present in both TLD and TLE) efficacy while reducing viral replicative fitness. Regarding DTG efficacy, during the same study period, we reported a prevalence of 0% resistance to DTG in ART naïve patients, inferring 100% predictive efficacy of DTG [17]. 

Subtyping was conducted using rapid subtyping tools such as COMET, REGA, and BLAST. Molecular phylogeny was used for subtype confirmation. Alignment and sequence cleaning was carried out using Aliview version 1.28, and phylogenetic tree inference was carried out using the maximum likelihood method on MEGA 11. The evolutionary history was inferred by using the Maximum Likelihood method and Tamura-Nei model. Initial tree(s) for the heuristic search were obtained automatically by applying Neighbor-Join and BioNJ algorithms to a matrix of pairwise distances estimated using the Tamura-Nei model and then selecting the topology with superior log likelihood value. Recombination events were evaluated using RDP.v3 and Simplot++ V1.3. Tree editing was carried out using iTOL version 6.7.3.

### 2.7. Data Entry and Statistical Analysis

The data was entered and analyzed using Epi Info version 7.2.2.6. Categorical variables were described in terms of frequency and percentage. The Chi-squared or Fischer’s exact tests were used to compare proportions of the qualitative variables. The values were expressed with their 95% confidence interval. The threshold of statistical significance was set at 5%. Any *p* < 0.2 from bivariate analysis was included for multivariate analysis through logistic regression, adjusting for potential confounders such as age, sex, region, or HIV-1 subtypes.

## 3. Results

### 3.1. General Characteristics

A total of 391 participants were recruited, with sequencing successfully carried out in 379 individual participant samples, giving a genotyping success rate of 96.9%. The 379 participants with successful sequencing data were from eight regions distributed as follows: 53 (14%) from the Center, 40 (10.6%) for the Far North, 46 (12.1%) for the East, 41 (10.8%) for the Littoral, 44 (11.6%) for the North, 61 (16.1%) for the Northwest, 49 (12.9%) for the West and 45 (11.9%) for the Southwest. The majority of participants were female (62%), with ages ranging from 20 to 80 years and a mean ± SD of 36 ± 10 years. The most represented age group was the 25 to 35 years age group (36%).

### 3.2. Genetic Diversity

Phylogenetic analysis of the sequences reveals that all the patients were infected with type 1 viruses, all belonging to group M. A total of 15 viral strains were identified. Recombinant forms (CRF02_AG, CRF11_cpx, CRF18_cpx, etc.) were the most represented, accounting for 81.1% of viral strains as compared to 18.2% for pure strains (A1, A3, D, G, F1, F2, J). Overall, CRF02_AG was the most represented viral clade, with 65.4% of the total workforce (see Figure 1). The complex forms identified were CRF04_cpx, CRF06_cpx, CRF09_cpx, CRF11_cpx, CRF18_cpx, CRF37_cpx, and CRF45_cpx).

### 3.3. Genotypic Resistance Profile

Of the 379 sequences obtained, 57 had at least one major resistance mutation, i.e., a prevalence overall pre-treatment resistance of 15% [95% CI: 11.8; 19]. At the regional level, we observed a significant disparity between the values of the global resistance threshold, ranging from 9.8% in the North region to 27.5% in the Far North region, *p* = 0.03.

Both nationally and regionally, the class of antiretrovirals with the highest resistance rate was that of NNRTIs. The national rate of resistance to NNRTIs, all generations combined, was 12.4% [95% CI: 9.5; 16.1]. It was highest in the Far North region, at 22.5% [95% CI 10.8; 38.4]. However, there was a non-significant disparity between the different regions, *p* = 0.16. Regarding resistance to first-generation NNRTIs, namely EFV and NVP, the national threshold was 7.9% [95% CI: 5.6; 11.1], with extremes of: 4.5% in the North to 15% in the Far North. Two out of eight regions crossed the 10% threshold; these are the regions of the East (10.9%) and the Far North (15%). We identified a total of five mutations responsible for resistance to these NNRTI. The most prevalent of them was the K103N mutation, identified in 21 (5.5% [95% CI: 3.7; 8.3]) of participants’ sequences. This was followed by the E138AGK (3.2% [95% CI: 1.8; 5.4]), V106I (2.9% [95% CI: 1.6; 5.1]), A98G (0.8% [95% CI: 0.3; 2.3]), and G190E (0.8% [95% CI: 0.3; 2.3]). As concerns NRTI, twelve sequences had at least one resistance mutation to this drug class, representing a national prevalence of 3.2% [95% CI: 1.8; 5.4]. The Southwest region was most affected, with a prevalence of 8.9% [95% CI: 2.5; 21.2]. Mutations responsible for drug resistance in this drug class were mainly the M184V mutation, responsible for resistance to 3TC, identified in six participants (1.6%, [95% CI: 0.7; 3.4]), followed by the K65R mutation, responsible for resistance to TDF, identified in four participants (1.1%, [95% CI: 0.3; 2.9]). Elsewhere, we identified thymidine analogue mutations responsible for resistance to AZT; K219REN, 4(1.1% [95% CI: 0.3; 2.9]) and M41L 2 (0.5% [95% CI: 0.09; 2.1]). Reverse transcriptase accessory mutations were also identified, notably the S68G mutation (4.0%, 95% CI: 2.4–6.4%). As concerns PI/r, overall pre-treatment resistance to PI/r was 1.3% [95% CI: 0.6; 3.1]. Pre-treatment PI/r resistance was only observed in three of the eight regions (Far North, East, and West), and was highest in the East region, 4.3% [95% CI: 0.5; 14.9]. The mutations responsible for PI/r resistance were M46I (0.3% [95% CI: 0.05; 1.5]), G73S (0.3%), V82F (0.3%), L89V (0.3%), and L90M (0.3%). The accessory mutations identified in the protease region included the L24F mutation (0.3%), L33F mutation (0.3%), K43T (0.3%) mutation, and the I47M (0.3%) mutation. Figure 2 shows the proportions of PDR by drug class, while Figure 3 shows the regional mapping of EFV/NVP- PDR by region covered during the survey.

There was no change in the trend of PDR mutations over time; *p* > 0.05. This could be due to the fact that the same ART regimens were used for the first- and second-line throughout the period of 2014–2019 (first-line: NNRTI-based; second-line: PI/r-based). The trends may change substantially with the transition to Dolutegravir-based regimens in the coming years.

### 3.4. Association between Pre-Treatment Drug Resistance, HIV-1 Genetic Diversity, Clinical and Demographics Parameters of the Study Participants

We found no significant associations between the presence of PDR and sex (*p* = 0.54), median age (*p* = 0.19), WHO clinical stage (*p* = 1), or even HIV-1 genetic diversity (*p* = see Table 2). Table 2 shows the occurrence of PDR with respect to pure versus recombinant strains, CRF02-AG versus non-CRF02-AG strains, and lastly, complex versus non-complex strains. We observed higher rates of PDR in participants from the urban regions (26.7%) as compared to the rural region (12.8%); OR = 2.4, *p* = 0.021. After multivariate analysis, adjusting for age, sex, and subtype, the region of residence (urban region) was independently associated with increased PDR (adjusted *p* = 0.016).

### 3.5. Predictive Efficacy of TLE versus TLD

By taking into account the resistance to NRTIs (and therefore the effectiveness of TDF) and to NNRTIs for the TDF + 3TC + EFV protocol (TLE) on the one hand and the efficacy of TDF for the TDF + 3TC + DTG protocol (TLD) on the other hand (DTG being totally effective a priori), we predicted the efficiency of the TDF + 3TC + DTG protocol of 98% against 92% for TDF + 3TC + EFV (*p* < 0.0001). This superiority of the TLD compared to TLE was also observed in all eight regions, including a region like the Far North, where the TLE protocol is only 85% effective compared to 100% TLD. Figure 4 shows the predictive efficacy of TLE as compared to TLD in the eight regions.

## 4. Discussion

Nationally, in all eight regions studied, we found a high pre-treatment resistance rate (15%), regardless of the class of antiretroviral molecules. However, there was a great disparity between the eight regions, with resistance rates varying from 6.8% in the North region to 26.1% and 27.5% in the East and Far North regions, respectively. A study conducted in Cameroon and published in 2018 found a similar prevalence, i.e., 14.2% of pre-treatment resistance in urban areas [14]. In Namibia and Sierra Leone, two other studies found a high prevalence of resistance in ART-naive patients, 12.7% and 36.7%, respectively [18,19]. These data are evidence of an increase in pre-treatment resistance in resource-constrained countries due to the expansion of access to antiretroviral therapy [11]. To add to this, this high rate of PDR observed is in the context of minimal risks of previous ARV exposure (following study inclusion criteria). It is therefore expected that for those with previous ART exposure (PMTCT, ART, PrEP, etc.), PDR may at least be similar, or even higher, as it has been shown that patients with previous ARV exposure have significantly higher rates of PDR as compared to those without any ARV exposure [8]. The disparity between these regions observed in our study could be explained by the difference in the performance of the control indicators, and therefore suggests that decision making should consider local realities. Similar to other studies, we observed higher rates of PDR in urban areas as compared to rural areas [14,20]. This is consistent with longer and more extensive use of ART in urban areas as compared to rural areas and, therefore, higher rates of PDR. Though this appears comforting for rural settings, it calls for continuous monitoring and preventive measures in these settings, as ART scale-up continues to be more effective over the years, extensively covering rural localities.

In terms of resistance according to the antiretroviral drug classes, NNRTIs had higher resistance rates (12.4%, including 7.9% resistance to first-generation NNRTIs, namely Efavirenz and Nevirapine) as compared to other ARV classes. Several other studies, both in sub-Saharan Africa and elsewhere, also found a high prevalence of pre-treatment drug resistance to NNRTIs, with the predominant mutation being the K103N, as also found in our study [14,18,19,21]. The reasons for this predominance of resistance to NNRTIs are multiple: the former use of NVP as monotherapy in the context of prevention of mother-to-child transmission in Cameroon as well as in many other countries; long-term use of EFV and/or NVP as the driving arm of first-line treatment protocols; the low genetic barrier of NNRTIs; and the ease of transmission of the K103N mutation and its good fitness. These are all elements that can explain this predominance of resistance to NNRTIs [22,23]. We observed a great disparity in the prevalence of resistance to first-generation NNRTIs between the regions, specifically the Far North and East regions presenting WHO’s critical threshold of 10%. This thus merits transition to DTG-based regimens or the use of genotypic resistance testing in choosing an optimal regimen for treatment initiation. This high prevalence of resistance to NNRTIs in these two regions could be explained by the social and security situation that prevails there, with the movements of populations coming from neighbouring countries that share borders with these regions. This suggests that the implementation of the WHO recommendations (transition to Dolutegravir for all patients on initiation of treatment) should be considered a priority in these regions. We found similar trends in PDR drug resistance mutations during the study period. This is similar to other studies, showing similar patterns of PDR essentially driven by NNRTI mutations, followed by NRTI mutations and, to a lesser extent, PI/r [14,20]. The similar patterns in these DRMs between 2014 and 2019 could be to the fact that the same ART regimens were used for the first- and second-line throughout the period of 2014–2019 (first-line: NNRTI-based; second-line: PI/r-based). Nonetheless, these trends may change substantially with the transition to Dolutegravir-based regimens in the coming years. Therefore, though current rates of PDR to integrase strand transfer inhibitor (INSTI) are low in our context (0% in INSTI naïve patients), a continuous update of this data is necessary for the long-term success of ART programs [17,24]. 

Genetic diversity results from recombination phenomena and errors that occurred during RNA reverse transcription due to the low level of fidelity of the reverse transcriptase enzyme. Phylogenetic analysis of the pol region sequences showed great genetic variability in the population, with a total of 18 viral strains, all belonging to HIV-1, with a predominance of the recombinant forms. The recombinant form CRF02_AG remains the predominant strain in our context, as demonstrated in several other studies [14,25]. The predominance of this subtype could be due to a biological advantage that this subtype has over the other subtypes, in particular, its great replicative capacity. We did not find any significant association between the viral strain and the pre-treatment resistance, thus suggesting that the initiation of antiretroviral treatment in our context could be conducted independently of the virus strain that the patient harbors. This great genetic diversity constitutes a real challenge for the diagnosis, treatment, and development of an HIV vaccine, the major part of diagnostic tools and treatment being designed on majority subtypes in developed countries, in particular subtype B [26].

Analysis of resistance mutations, in particular the determination of viral susceptibility scores for each antiretroviral molecule, made it possible to compare the effectiveness of two standard first-line protocols. One of these protocols, TDF + 3TC + EFV, was the preferred first-line protocol used up until 2019 in Cameroon, and the other, TDF + 3TC + DTG, was recommended recently by the WHO, given evidence of increasing levels of pre-treatment drug resistance to NNRTIs. Our findings show a predicted significant superiority of the TDF + 3TC + DTG protocol, both nationally and regionally. This observation reinforces the idea that Dolutegravir remains the molecule of choice for first-line protocols and thus suggests an acceleration of the transition process, strongly recommended by the WHO. However, in addition to its efficacy, Dolutegravir is a controversial molecule due to some described side effects. Preliminary findings from the Tsepamo study suggested an increased risk of neural tube defects in children of pregnant women receiving DTG, with other studies reporting increased weight gain with DTG use [27,28,29,30]. However, recent data upon completion of the Tsepamo study refuted this link between DTG and neural tube defects [31,32]. Nonetheless, in this context of transition to first-line protocols based on Dolutegravir, implementing a pharmacovigilance mechanism within our management program remains crucial to better understand the risks of its use in real conditions.

Although this study provides us with an overview of pre-treatment HIV drug resistance in Cameroon, it nevertheless has certain limits, the main ones being that the technique used for carrying out resistance tests only detected a resistance mutation if it represented at least 20% of the circulating population. Thus, the real prevalence of resistance would be better evaluated if we had a high-throughput sequencing technique (1% detection threshold), a platform unavailable in our context. Elsewhere, with the Test and Treat strategy, CD4 T lymphocyte levels and baseline viral loads are unavailable at the time of ART initiation. This data would have made it possible to assess the resistance profile as a function of the patient’s immune status and viremia, which could have been important for a better description of this population. Finally, the lack of data from two of the ten regions of Cameroon, namely, the South and Adamawa, somewhat undermines the power of the study.

## 5. Conclusions

In Cameroon, the level of EFV/NVP PDR suggests a superior efficacy of TLD over TLE as an initial ART regimen. Moreover, the significant disparities in PDR levels according to regional settings is a call for urgent interventions within regions with PDR above the critical threshold (Far North and East) and for stratified monitoring of TLD effectiveness. Current evidence suggests interventions should be regardless of gender and viral strains in RLS.

## Figures and Tables

**Figure 1 viruses-15-01458-f001:**
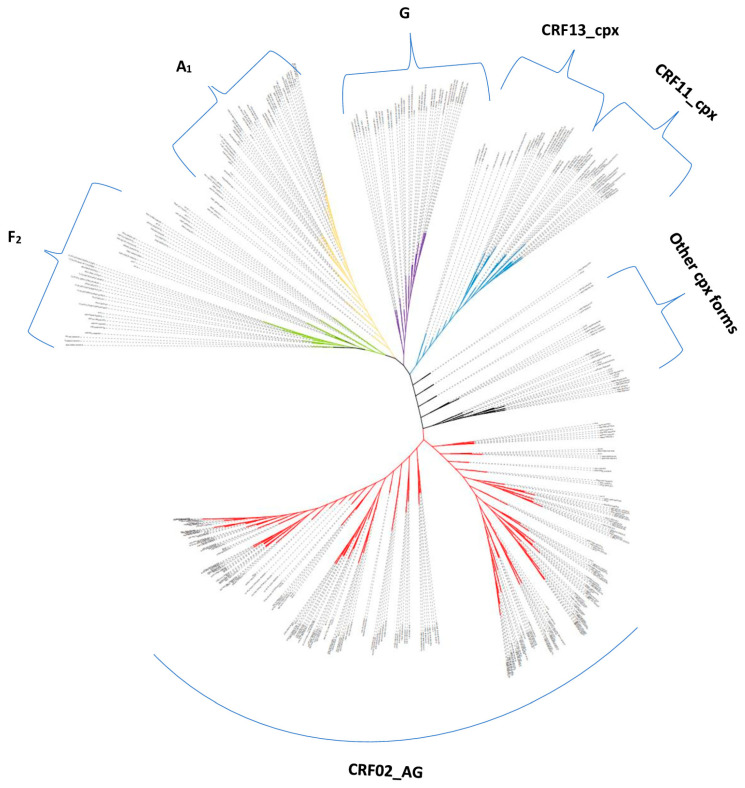
Phylogenetic Tree of HIV-1 sequences from ART-naive individuals. The evolutionary history was inferred by using the Maximum Likelihood method and Tamura-Nei model [1]. The tree with the highest log likelihood (−69,062.29) is shown. Initial tree(s) for the heuristic search were obtained automatically by applying Neighbor-Join and BioNJ algorithms to a matrix of pairwise distances estimated using the Tamura-Nei model and then selecting the topology with superior log. Cpx = complex recombinant forms.

**Figure 2 viruses-15-01458-f002:**
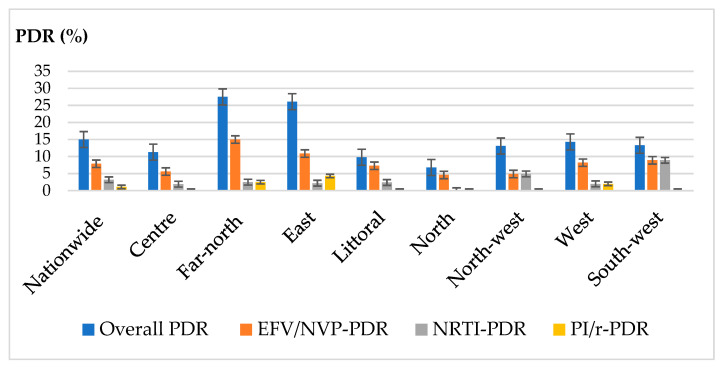
Pre-treatment drug resistance (PDR) according to ARV class nationally and by geographical region. The histogram shows rates of PDR by ARV class, both at the national level and across regions. We note that overall PDR is high, with two regions (Far North and East) surpassing the 10% threshold for EFV/NVP.

**Figure 3 viruses-15-01458-f003:**
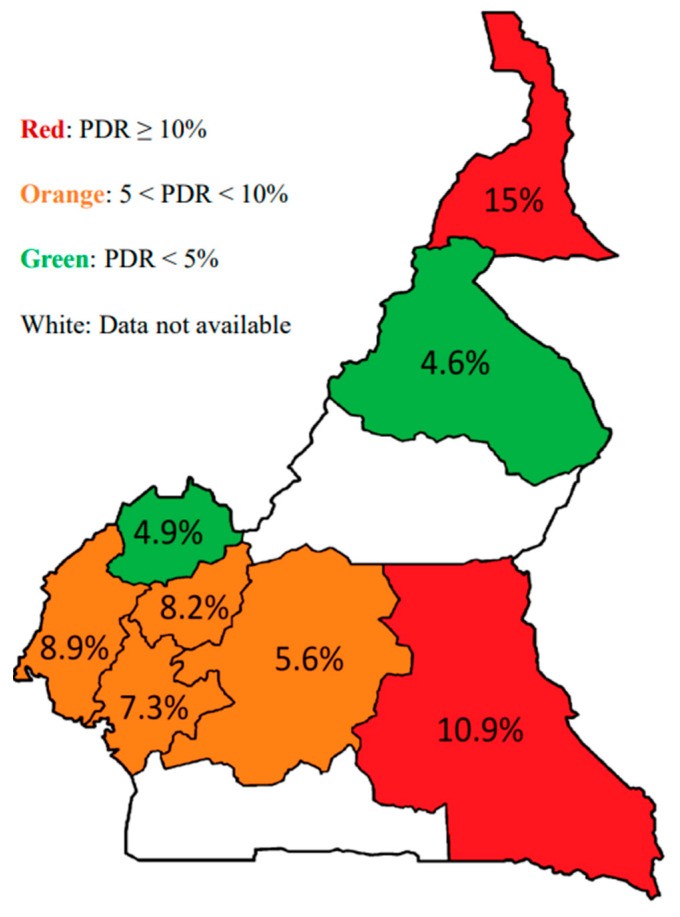
Regional mapping of EFV/NVP-PDR.The figure shows the mapping of EFV/NVP PDR. Red colour denotes regions with high levels of PDR (above the 10% threshold), orange denotes regions with moderate levels of PDR (5–10%), and green denotes regions with levels of PDR below 5%.

**Figure 4 viruses-15-01458-f004:**
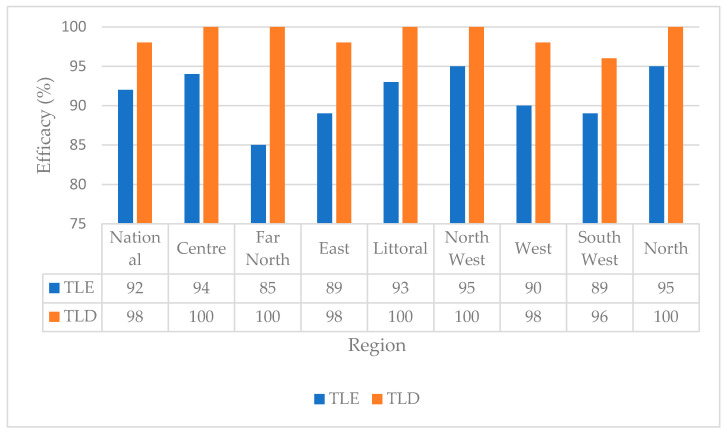
Predictive efficacy of TLE (TDF + 3TC + EFV) vs. TLD (TDF + 3TC + DTG) in Cameroon. The figure shows predictive efficacy of TLD as compared to TLE. We observe superior predictive efficacy nationally and across all regions, with a marked superiority in the Far North region where EFV/NVP pre-treatment drug resistance was highest.

**Table 1 viruses-15-01458-t001:** Recruitment sites and health facilities.

Region	Health Facility
Center	Yaoundé Central Hospital ^1^, Mbalmayo District Hospital ^2^
Littoral	Bonassama Hospital ^1^, Edea District Hospital ^2^

West	Bafoussam Regional Hospital ^1^, MIFI District Hospital ^1^, Foumban District Hospital ^2^, Bangou District Hospital ^2^
Northwest	Bamenda Regional Hospital ^1^, Mbingo Baptist Hospital ^2^
Southwest	Buea Regional Hospital ^1^, Mount Mary Hospital Buea ^1^, Buea Road Integrated Health Center ^1^, Ekona Medicalized Health Center ^2^, Muyuka District Hospital ^2^
North	Garoua Regional Hospital ^1^, Guider District Hospital ^2^
Far North	Maroua Regional Hospital ^1^, Kolofata District Hospital ^2^

NB: ^1^ Refers to a health facility in an urban site. ^2^ Refers to a health facility in a rural site.

**Table 2 viruses-15-01458-t002:** Effect of HIV-1 genetic diversity of PDR.

Variable	Subtype	Proportion with PDR (%)	OR	*p* Value
CRF02_AG vs. Non-CRF02_AG clades	CRF02-AG (N = 38)	16.0	0.8	0.4
	Non CRF02-AG (N = 26)	19.6		
Pure vs. recombinant clades	Pure Clades (N = 15)	17.4	1	1.0
	Recombinant (N = 49)	17.2		
Complex clades vs. other clades	Complex recombinants (N = 10)	23.3	1.5	0.3
	Others (N = 54)	16.5		

## Data Availability

Sequence data are available on GenBank under the accession numbers GenBank MK702015-MK702057, MK867695-MK867757, and MK995397-MK995457.

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
