# Peer review of "Pre-Treatment HIV Drug Resistance and Genetic Diversity in Cameroon: Implications for First-Line Regimens"

_viruses, 2023, doi:10.3390/v15071458_

Round 1

Reviewer 1 Report

In the manuscript “Pre-treatment HIV Drug Resistance and Genetic Diversity in

Cameroon: Implications for first-line regimens”, the authors reported a cross-sectional, HIV molecular epidemiology study describing HIV genotypes and drug resistance mutations in ART naïve populations in Cameroon. This study has its regional important for HIV surveillance, prevention, and treatment. Particularly, the authors estimated what percentage of the population would be susceptible to the new DTG-based regimen. This is an important analysis before rolling out the DTG-based regimen. I have several comments and suggestions to the authors.

1. The English can be improved and a few types need to be corrected. For instance, line 123, “where” should be “were”. Line 130-133. This sentence is hard to read.

2. In the methods section, there are duplicated description of study sites and study population. The subsection of “Study Population” and “Study Sites” need to be merged with the section of “Study design and population”.

3. The authors described the multivariable analysis in the methods section, but not used this analysis in the report. One thing I can suggest is that use this analysis to study the factors associated with drug resistance. The authors already performed univariable analysis. It will be interesting to see what the multivariable analysis would say.

4. HIV drug resistance mutations change over the time because the HIV epidemic changes. The samples were collected from 2014 to 2019. Has the authors noticed any year-by-year differences in the drug resistance mutations? Could the regional differences in drug resistance come from the bias in the sample collection in different years?

5. Table 2 is hard to read. The authors did three comparisons but did not separate them in the table.

6. Figure 2. Negative value of PDR does not make sense. Please start the y axis from 0, instead of -5.

7. Line 124. Why people with hepatitis B or hepatitis C were excluded?

8. Line 134. I assume only the plasma were stored in cryotubes. Please confirm.  

9. Line 377-378. It is still datable that DTG would cause neural tube defects in fetus. It is ok to discuss it but references are needed here. Ref 24 only described the side effect of obesity of DTG.

The English can be improved and a few types need to be corrected. For instance, line 123, “where” should be “were”. Line 130-133. This sentence is hard to read.

Author Response

Comments and Suggestions for Authors

In the manuscript “Pre-treatment HIV Drug Resistance and Genetic Diversity in

Cameroon: Implications for first-line regimens”, the authors reported a cross-sectional, HIV molecular epidemiology study describing HIV genotypes and drug resistance mutations in ART naïve populations in Cameroon. This study has its regional important for HIV surveillance, prevention, and treatment. Particularly, the authors estimated what percentage of the population would be susceptible to the new DTG-based regimen. This is an important analysis before rolling out the DTG-based regimen. I have several comments and suggestions to the authors.

  1. The English can be improved and a few types need to be corrected. For instance, line 123, “where” should be “were”. Line 130-133. This sentence is hard to read.

Response: We have revised lines 123 and 130-133 for better clarity as recommended in the revised manuscript (see in current version, line 132 to 133). We have also had fluent English-speaking co-authors go through the manuscript for an improvement of the English as requested. We hope at the current state, the English is much improved and easy to read and comprehend for any potential readers.

  1. In the methods section, there are duplicated description of study sites and study population. The subsection of “Study Population” and “Study Sites” need to be merged with the section of “Study design and population”.

Response: We have merged these sections accordingly in the first paragraph of the results section “Study design, sites and population.” (see lines 116 to 127).

  1. The authors described the multivariable analysis in the methods section, but not used this analysis in the report. One thing I can suggest is that use this analysis to study the factors associated with drug resistance. The authors already performed univariable analysis. It will be interesting to see what the multivariable analysis would say.

Response: We thank the reviewer for this remark. Nonetheless, we included a multivariate analysis in the results sections in lines 286-288 which reads “even after adjusting for age, sex and subtype (adjusted p=0.01)”. We admit it may have been missed due to a not very clear presentation. We have rephrased this sentence for better visibility as follows “After multivariate analysis, adjusting for age, sex and subtype, region of residence (urban region) was independently associated with increased PDR (adjusted p=0.016)” (see lines 292-294 in revised manuscript).  We have also enriched the discussion with respect to this result. (See line 344 to 349 in revised manuscript)

  1. HIV drug resistance mutations change over the time because the HIV epidemic changes. The samples were collected from 2014 to 2019. Has the authors noticed any year-by-year differences in the drug resistance mutations? Could the regional differences in drug resistance come from the bias in the sample collection in different years?

Response: We thank the reviewer for this remark. In fact, this analysis was performed, and there was no change in the trend of PDR mutations overtime; this could be to the fact that the same ART regimens were used for first- and second-line throughout the period of 2014-2019 (first-line: NNRTI-based; and second-line: PI/r-based); p>0.05. The trends may change substantially with transition to dolutegravir-based regimens in the coming years. We have added this in the results (lines 277 to 280) and discussion (371 to 380).

  1. Table 2 is hard to read. The authors did three comparisons but did not separate them in the table.

Response: We have re-edited table 2 and separated these comparisons for better comprehension. We hope it is more understandable for any other potential readers in its current form.

  1. Figure 2. Negative value of PDR does not make sense. Please start the y axis from 0, instead of -5.

Response: We have corrected this figure as recommended.

  1. Line 124. Why people with hepatitis B or hepatitis C were excluded?

Response: We excluded patients with viral hepatitis as this could be a potential confounder towards occurrence of PDR. For example, some antiretroviral (e.g 3TC, TDF, TAF) have activity against both HIV and viral hepatitis, thus previous exposure to these drugs in cases of viral hepatitis could have impact on HIV PDR selection. Furthermore, studies have shown increased rates of HBV mutations in HIV/HBV co-infected patients, and differences in Hepatitis C genotypes in HCV mono-infected patients as compared to HCV/HIV co-infected patients. Therefore, apart from the possibility of drug exposure due to co-infection, the interaction between these viruses might also influence occurrence of HIV PDR ( https://doi.org/10.1002/jmv.25078)

A concise statement has been added as an explanatory note to justify the exclusion in the text, as follows: Patients co-infected with viral hepatitis were to excluded to limit confounders related to potential HBV drug exposure that interferes with HIV PDR selection (tenofovir, lamivudine, emtricitabine); HCV/HIV co-infection also exhibits differences in viral genotypes as compared to mono-infection (lines 133 to 137).

  1. Line 134. I assume only the plasma were stored in cryotubes. Please confirm.

Response: We confirm this and have corrected the sentence in line 134 for better clarity (see lines 146 and 147 in the revised manuscript).

  1. Line 377-378. It is still datable that DTG would cause neural tube defects in foetus. It is ok to discuss it but references are needed here. Ref 24 only described the side effect of obesity of DTG.

Response: We thank and agree with the reviewer on this remark. We also agree and are aware that recent data refutes the initial attribution of DTG-use to the occurrence of neural tube defects. We have revised this part of the discussion, while also adding and updating relevant bibliographic references. Please find in the revised manuscript, a much-improved description of this section (Lines 404 to 412).  

Comments on the Quality of English Language

The English can be improved and a few types need to be corrected. For instance, line 123, “where” should be “were”. Line 130-133. This sentence is hard to read.

Response: We thank the reviewer for this remark. We have corrected these in the manuscript and gone through the entire manuscript for better English editing.

Reviewer 2 Report

This is a very well designed and well written study addressing an important topic. I have several minor comments.

1. The authors cite several previous studies of PDR in Sub-Saharan Africa. However, the introduction and discussion and the overall context of the study would be improved if they focused specifically on the analyses in the 2021 WHO report on HIV drug resistance (https://apps.who.int/iris/handle/10665/349340) and on previous studies from Cameroon.

2. If the authors have data on which patients with PDR received previous therapy (i.e., they were re-initiating first-line therapy) that information would be useful.

3. In several places the authors compare the prevalence of PDR between two different groups (e.g., between different regions or subtypes). However, considering that multiple groups were examined, it would be more appropriate to ask whether the distribution of PDR across different populations was non-random rather than just comparing groups with the highest and lowest PDR prevalences.

4. It is unclear why the authors use the abbreviation TELE rather than TLE.

Author Response

This is a very well designed and well written study addressing an important topic. I have several minor comments.

  1. The authors cite several previous studies of PDR in Sub-Saharan Africa. However, the introduction and discussion and the overall context of the study would be improved if they focused specifically on the analyses in the 2021 WHO report on HIV drug resistance (https://apps.who.int/iris/handle/10665/349340) and on previous studies from Cameroon.

Response: We thank the reviewer for this remark. We have used this reference to improve our manuscript as recommended. We have also added some more studies discussed more on some studies carried out in Cameroon. In the revised manuscript, you can find updates in the introduction (line 89 to 92), and the discussion sections. (337 to 342).

  1. If the authors have data on which patients with PDR received previous therapy (i.e., they were re-initiating first-line therapy) that information would be useful.

Response: We thank the reviewer for this comment. Since our focus was essentially on patients initiating ART for the first time (representing the majority of patients in our programmatic setting), we did not include patients who reported prior exposure to ART (they were also fewer cases identified). Our results therefore describe PDR levels in a population of first-ART initiators, hence implying that evidence-based recommendations are tailored toward this target population. It is expected that for those with previous exposure to ART (PMTCT, ART, PrEP, etc), PDR-levels may be at least similar or even higher. We have given more insight in the discussion, to highlight this comment. (lines 331 to 334)

  1. In several places the authors compare the prevalence of PDR between two different groups (e.g., between different regions or subtypes). However, considering that multiple groups were examined, it would be more appropriate to ask whether the distribution of PDR across different populations was non-random rather than just comparing groups with the highest and lowest PDR prevalence.

Response: We thank the reviewer for this remark. We also agree that adjusting for confounders is critical when comparing the level of PDR in different populations. To address this, we carried out a multivariate analysis including all potential confounders and still found that those in the urban areas had greater odds of experiencing PDR. Please find this in the results section (lines 301 to 303) and in the discussion (lines 345 to 350).

  1. It is unclear why the authors use the abbreviation TELE rather than TLE.

Response: Locally, TELE has been frequently used to describe the TDF+3TC+EFV combination. For consistency with international abbreviation of this regimen, we agree with the reviewer that TLD is more standard and have thus changed TELE to TLE although the manuscript.

Reviewer 3 Report

In this study the authors present the results of the pre-treatment HIV drug resistance analysis in eight regions of Cameroon and its implications on first-line regimens. The study was conducted between 2014 and 2019 and included 379 HIV-positive persons. 

For the patients included the HIV-1 pol gene was sequenceby Sanger and analyzed for the mutations, subtypes and recombinants forms of HIV. A very large genetic diversity of HIV viruses and a significant predominance of recombinant variants, including complex ones, was shown. A thorough analysis of individual mutations and classes of inhibitors was performed.

Data are presented illustrating the low treatment coverage, insufficient treatment success rate and, as a result, a high prevalence of resistance mutations. The predominant prevalence of mutations to NNRTIs was shown, indicating the need for the urgent need in the transition to dolutegravir-containing treatment regimens. An important piece of information concerns the uneven distribution of the level of HIV drug resistance among the regions of the country.

In fact, the article contains all the necessary and sufficient information on the presented work and the correct conclusions. I find it very interesting and useful. There are only minor remarks, mostly typos, which are marked in the attached file.

Comments

Line 93-94.  “The threshold for HIV pre-treatment resistance to ARVs is generally increasing in low-income countries with annual rates of increase reaching 29% in East Africa”. Do the authors really mean an annual increase, or 29% points to the achieved prevalence of resistance?

Line 122-123.  “Not included in our study where patients … whose samples were unsuccessful at sequencing…”. Wouldn't it be better to point to the number of patients included in the study, and then the number of those in whom sequencing was successful?

Line 133-134.  “The tubes were centrifuged at a speed of 1800 rpm, aliquoted (1 ml) in cryotubes…”. It should be indicated that blood plasma but not tubes were aliquoted.

Figure 1. It would be worthwhile to somehow designate the branches marked in black (complex recombinants?).

Line 256. A96G – is it really associated with NNRTI resistance?

Table 2. I recommend dividing it into three parts according to comparisons - for better perception by readers.

no comments

Author Response

Comments and Suggestions for Authors

In this study the authors present the results of the pre-treatment HIV drug resistance analysis in eight regions of Cameroon and its implications on first-line regimens. The study was conducted between 2014 and 2019 and included 379 HIV-positive persons. 

For the patients included the HIV-1 pol gene was sequenced by Sanger and analyzed for the mutations, subtypes and recombinants forms of HIV. A very large genetic diversity of HIV viruses and a significant predominance of recombinant variants, including complex ones, was shown. A thorough analysis of individual mutations and classes of inhibitors was performed.

Data are presented illustrating the low treatment coverage, insufficient treatment success rate and, as a result, a high prevalence of resistance mutations. The predominant prevalence of mutations to NNRTIs was shown, indicating the need for the urgent need in the transition to dolutegravir-containing treatment regimens. An important piece of information concerns the uneven distribution of the level of HIV drug resistance among the regions of the country.

In fact, the article contains all the necessary and sufficient information on the presented work and the correct conclusions. I find it very interesting and useful. There are only minor remarks, mostly typos, which are marked in the attached file.

Comments

Line 93-94.  “The threshold for HIV pre-treatment resistance to ARVs is generally increasing in low-income countries with annual rates of increase reaching 29% in East Africa”. Do the authors really mean an annual increase, or 29% points to the achieved prevalence of resistance?

Response: We thank the reviewer for this remark.  We agree that this sounds confusing indeed. The authors referenced here (Gupta et al) carried out a meta-regression model, where they described risks of about 30% increase of PDR per year in east African resource limited settings. To avoid confusion, and enable better contextualization of data, we have rephrased this sentence and hope it is more comprehensive for any potential readers (line 98 to 100).

Line 122-123.  “Not included in our study where patients … whose samples were unsuccessful at sequencing…”. Wouldn't it be better to point to the number of patients included in the study, and then the number of those in whom sequencing was successful?

Response: We agree with the reviewer on this remark. To ensure adequate statistics (use of correct denominator for calculation of PDR proportions), we were obliged to exclude samples which were not successfully sequenced. However, we agree that the number of excluded participants due to failed sequencing merits mentioned in the results section. In all, we had 391 participants, and successfully     amplified/sequenced 379. We have added this to the results section (Lines 219 to 221).

Line 133-134.  “The tubes were centrifuged at a speed of 1800 rpm, aliquoted (1 ml) in cryotubes…”. It should be indicated that blood plasma but not tubes were aliquoted.

Response: We have updated this accordingly (lines 146 to 147 in revised manuscript)

Figure 1. It would be worthwhile to somehow designate the branches marked in black (complex recombinants?).

Response: We thank the reviewer for this comment. Indeed, the said branches represent the other viral clades (mostly complex forms), observed at lower frequencies. We have updated figure 1 accordingly in the revised manuscript.

Line 256. A96G – is it really associated with NNRTI resistance?

Response: We agree with the reviewer as this is a typo error. We meant to write A98G, which is an NNRTI mutation. We have corrected in the revised manuscript (Line 251).

Table 2. I recommend dividing it into three parts according to comparisons - for better perception by readers.

Response: We have revised this table as requested for better comprehension.

Round 2

Reviewer 1 Report

The authors have addressed my previous comments.